# Fast Slate Policy Optimization: Going Beyond Plackett-Luce

**Otmane Sakhi**
*CREST-ENSAE, Criteo AI Lab*                                           *o.sakhi@criteo.com*

**David Rohde**
*Criteo AI Lab*                                                       *d.rohde@criteo.com*

**Nicolas Chopin**
*CREST-ENSAE*                                               *nicolas.chopin@ensae.fr*

**Reviewed on OpenReview:** *https://openreview.net/forum?id=f7a8XCRtUu*

## Abstract

An increasingly important building block of large scale machine learning systems is based on returning *slates*; an ordered lists of items given a query. Applications of this technology include: search, information retrieval and recommender systems. When the action space is large, decision systems are restricted to a particular structure to complete online queries quickly. This paper addresses the optimization of these large scale decision systems given an arbitrary reward function. We cast this learning problem in a policy optimization framework and propose a new class of policies, born from a novel relaxation of decision functions. This results in a simple, yet efficient learning algorithm that scales to massive action spaces. We compare our method to the commonly adopted Plackett-Luce policy class and demonstrate the effectiveness of our approach on problems with action space sizes in the order of millions.

## 1 INTRODUCTION

Large scale online decision systems, ranging from search engines to recommender systems, are constantly queried to deliver ordered lists of content given contextual information. As an example, a user who has just seen the film 'Batman', might be recommended: Superman, Batman Returns, Bird Man, or a user that is reading a page about 'technology' might be interested in other stories about biotechnology, solar power, and large language models. A large scale production system must therefore be able to rapidly respond to a query like 'Batman' or 'technology' with an ordered list of relevant items.

A proven solution to this problem is to generate the ordered list using an approximate maximum inner product search (MIPS) algorithm (Shrivastava & Li, 2014), which, at the expense of a constraint in the decision rule, provides extremely rapid querying even for massive catalogues. While MIPS based systems are a proven technology at deployment time, the offline optimization of them is not, and the standard algorithm based on policy learning using the Plackett-Luce distribution is infeasibly slow, as the algorithm both iterates slowly and requires very large numbers of iterations to converge. Fortunately, other approaches are possible which both iterate faster and require fewer iterations. In this paper, we propose a simple algorithm that enjoys better convergence properties than competitors.

We denote by $x \in \mathcal{X}$ a context; it can be the history of the user, a search query or even a whole web page. The decision system is tasked to deliver, given the context $x$, an ordered list of actions $A_K = [a_1, ..., a_K]$ of arbitrary size $K \in \mathbb{N}^*$, often referred to as *slates*. This slate can be an ad banner, a list of recommended content or search results. Our decision system, given contextual information $x$, constructs a slate by selecting a subset of actions $\{a_1, ..., a_K\} \subset \mathcal{A}$ from a potentially large discrete set $\mathcal{A}$ and ordering them. Let $P = |\mathcal{A}|$ be the size of the action set. Fixing the slate size $K$, we model our system by a decision function $d : \mathcal{X} \rightarrow \mathcal{S}_K(\mathcal{A})$ that maps contexts $x$ to the space $\mathcal{S}_K(\mathcal{A})$ of ordered lists of size $K$. Each pair of context $x$ and slate $A_K$ is

associated with a reward function[1] $r(A_K, x)$ that encodes the relevance of the slate $A_K$ for $x$. We suppose that the contexts are stochastic, and coming from an unknown distribution $\nu(\mathcal{X})$. Our objective is to find decision systems that maximize the expected reward, given by:

$$\mathbb{E}_{x \sim \nu(\mathcal{X})} \left[ r(A_K = d(x), x) \right].$$

Assuming that we have access to the reward function, the solution of this optimization problem is given by:

$$\forall x \in \mathcal{X}, \quad d(x) = \underset{A_K \in \mathcal{S}_K(\mathcal{A})}{\arg\max} \; r(A_K, x). \tag{1}$$

This solution, while optimal, is intractable as it requires, for a given context $x$, the search in the enormous space $\mathcal{S}_K(\mathcal{A})$ of size $\mathcal{O}(P^K)$ to find the best slate. Instead of maximizing the expected reward over the whole space, we want to restrict ourselves to decision functions of practical use. In this direction, we begin by defining the parametric relevance function $f_\theta : \mathcal{A} \times \mathcal{X} \to \mathbb{R}$:

$$\forall (a, x) \in \mathcal{A} \times \mathcal{X}, \quad f_\theta(a, x) = h_\Xi(x)^T \beta_a,$$

with the learnable parameter $\theta = [\Xi, \beta]$, a parametric transform $h_\Xi : \mathcal{X} \to \mathbb{R}^L$ that creates a context embedding of size $L$, and $\beta_a$ the embedding of action $a$ in $\mathbb{R}^L$. the embedding dimension $L$ is usually taken to be much smaller than $P$, the size of the action space. We then define our decision function:

$$\forall x \in \mathcal{X}, \quad d_\theta(x) = \underset{a \in \mathcal{A}}{\mathrm{argsort}^K} \left\{ h_\Xi(x)^T \beta_a \right\}. \tag{2}$$

with $\mathrm{argsort}^K$ the argsort function truncated at the $K$-th item. Restricting the space of decision functions to this parametric form reduces the complexity of returning a slate for the context $x$ from $\mathcal{O}(P^K)$ to $\mathcal{O}(P \log K)$. This complexity can be further decreased. Equation (2) transforms the querying problem to the MIPS: Maximum Inner Product Search problem (Shrivastava & Li, 2014), for which different algorithms were proposed (Gionis et al., 1999; Malkov & Yashunin, 2020; Guo et al., 2020) to find a solution in a logarithmic time complexity. These algorithms build fixed indexes over the action embeddings $\beta$, with particular structures to allow the identification (sometimes approximate) of the $K$ actions with the largest inner product with the query $h_\Xi(x)$. This allow us to reduce even further the complexity of the $\mathrm{argsort}^K$ operator from $\mathcal{O}(P \log K)$ to a logarithmic time complexity $\mathcal{O}(\log P)$, making fast decisions possible in problems with massive action spaces $\mathcal{A}$. This leaves us with the problem of finding the optimal decision function within the constraints of this parametric form. This is achieved by solving the following:

$$\theta^* \in \underset{\theta = [\Xi, \beta]}{\arg\max} \; \mathbb{E}_{x \sim \nu(\mathcal{X})} \left[ r(A_K = d_\theta(x), x) \right].$$

As we do not have access to $\nu(\mathcal{X})$, we replace the previous objective with its empirical counterpart:

$$\theta^* \in \underset{\theta = [\Xi, \beta]}{\arg\max} \; \frac{1}{N} \sum_{i=1}^{N} \hat{r} \left( A_K = d_\theta(x_i), x_i \right), \tag{3}$$

with $\{x_i\}_{i \in [N]}$ observed contexts and $\hat{r}$ an offline reward estimator; it includes the Direct Method, Inverse Propensity Scoring (Horvitz & Thompson, 1952), Doubly Robust Estimator (Dudík et al., 2014) and many other variants, as presented in (Sakhi et al., 2023b). The optimization problem of Equation (3) is complicated by the fact that the reward can be non-smooth and that our decision function is not differentiable. A way to handle this is by relaxing the optimization objective. Differentiable sorting algorithms (Grover et al., 2019; Prillo & Eisenschlos, 2020) address a similar problem but make strong assumptions about the structure of the reward function, and cannot scale to large action space problems. To be as general as possible, we take another direction and relax the problem into an offline policy learning formulation (Bottou et al., 2013; Swaminathan & Joachims, 2015). We extend our space of parametrized decision functions to a well chosen space of stochastic policies $\pi_\theta : \mathcal{X} \to \mathcal{P}(\mathcal{S}_K(\mathcal{A}))$, that given a context $x$, define a probability distribution

---

[1]motivated by business metrics and/or users engagement.

| | Arbitrary Reward | Low Gradient Variance | Time Complexity | Space Complexity |
|---|:---:|:---:|:---:|:---:|
| `PL-PG` | ✓ | ✗ | $\mathcal{O}(SP)$ | $\mathcal{O}(SP)$ |
| `PL-Rank` | ✗ | ✓ | $\mathcal{O}(SP)$ | $\mathcal{O}(SP)$ |
| `LGP` | ✓ | ✓ | $\mathcal{O}(SP)$ | $\boldsymbol{\mathcal{O}(SL)}$ |
| `LGP-MIPS` | ✓ | ✓ | $\boldsymbol{\mathcal{O}(S \log P)}$ | $\boldsymbol{\mathcal{O}(SL)}$ |

**Table 1:** High level comparison between the different optimization algorithms for slate decision functions, with $P$ the size of the action space, $L$ the size of the embedding space and $S$ the number of samples used to approximate the gradient. `PL-PG` is Plackett-Luce trained with the Score Function Gradient (Williams, 1992). `PL-Rank` is the algorithm proposed in Oosterhuis (2022). `LGP` is our proposed method and `LGP-MIPS` is its accelerated variant. Our method works with arbitrary rewards, scales logarithmically with $P$ and have low memory footprint ($L \ll P$).

over the space of slates of size $K$. Given a policy $\pi_\theta$, we relax Equation (3), taking an additional expectation under $\pi_\theta$ to obtain:

$$\theta^* \in \underset{\theta=[\Xi,\beta]}{\arg\max} \hat{R}(\pi_\theta) = \frac{1}{N} \sum_{i=1}^{N} \mathbb{E}_{A_K \sim \pi_\theta(\cdot|x_i)} \left[ \hat{r}(A_K, x_i) \right]. \tag{4}$$

The most common policy class for this type of problem is Plackett-Luce (Plackett, 1975), that generalises the softmax parametrization to slates of size $K > 1$. Under this policy class, computing exact gradients is intractable but we can obtain approximate gradients w.r.t to $\theta$ of Equation (4). Common approximations (Williams, 1992) are based on sampling from the policy, are computed in $\mathcal{O}(P \log K)$ and suffer from a variance that grows with the slate size $K$. In the special case of decomposable rewards over items on the slate (Swaminathan et al., 2017), exploiting this linearity structure (Oosterhuis, 2022) provides gradient estimates with better variance. However, training speed still scales linearly with $P$ making policy learning infeasible in large action spaces.

In this work, we propose **LGP: Latent Gaussian Perturbation**, a new policy class based on smoothing the latent space, that is perfectly suitable to optimize decision functions of the form described in (2). As shown in Table 1, our method provides fast sampling, low memory usage, gradient estimates with better computational and statistical properties while being agnostic to the reward structure. When the embeddings are prefixed, this class naturally benefits from approximate MIPS technology, making sampling logarithmic in the action space size and opening the possibility for policy optimization over billion-scale space sizes.

This paper will be structured as follows. In Section 2, we will review the Plackett-Luce policy class and present its limitations. Section 3 will introduce our newly proposed relaxation, motivate its use and propose a learning algorithm. We focus in Section 4 on experiments to validate our findings empirically. Section 5 will cover the related work and we conclude with Section 6.

## 2 Plackett-Luce Policies

### 2.1 A Simple Definition

We relax our objective function and model online decision systems as stochastic policies over the space of slates; ordered lists of actions. There are some natural parametric forms to define a policy on discrete action spaces. If we are dealing with the simple case of $K = 1$ (slate of one action), we can adopt the softmax policy (Swaminathan & Joachims, 2015; Sakhi et al., 2023b) that, conditioned on the context $x \in \mathcal{X}$ and for a particular action $a \in \mathcal{A}$, is of the form :

$$\pi_\theta(a|x) = \frac{\exp\{f_\theta(a,x)\}}{\sum_b \exp\{f_\theta(b,x)\}} = \frac{\exp\{f_\theta(a,x)\}}{Z_\theta(x)}.$$

This softmax parametrization found great success (Swaminathan & Joachims, 2015; Chen et al., 2019; Sakhi et al., 2023b), and is ubiquitous in applications where the goal is to learn policies or distributions over discrete actions. Once we deal with $K > 1$, one can generalize the previous form giving us the Plackett-Luce

policy (Plackett, 1975). For a given $x$ and a particular slate $A_K = [a_1, ..., a_K]$, we write its probability:

$$\pi_\theta(A_K|x) = \prod_{i=1}^{K} \frac{\exp\{f_\theta(a_i, x)\}}{Z_\theta^{i-1}(x)} = \prod_{i=1}^{K} \pi_\theta(a_i|x, A_{1:i-1}), \tag{5}$$

with $Z_\theta^0(x) = Z_\theta(x)$ and $Z_\theta^i(x) = Z_\theta^{i-1}(x) - \exp\{f_\theta(a_i, x)\}$.

Computing these probabilities can be done in $\mathcal{O}(P)$ which is comparable to the simple case where $K = 1$. The probabilities given by the Plackett-Luce policy are intuitive. Equation (5) can be seen as the probability to generate the slate $A_K = [a_1, ..., a_K]$ by sampling without replacement from a categorical distribution over the discrete space $\mathcal{A}$ with action probabilities proportional to $\exp\{f_\theta(a, x)\}$. This sampling procedure can be done in $\mathcal{O}(KP)$, but its sequential nature is a bottleneck for parallelization. Another way to sample from this distribution is to exploit the following expression:

$$\pi_\theta(A_K|x) = \mathbb{E}_{\gamma \sim \mathcal{G}^P(0,1)} \left[ \mathbb{1} \left[ A_K = \underset{a' \in \mathcal{A}}{\text{argsort}}^K \{f_\theta(a', x) + \gamma_i\} \right] \right],$$

with $\gamma \sim \mathcal{G}^P(0, 1)$ a vector of $P$ independent Gumbel random variables. This is known in the literature as the Gumbel trick (Huijben et al., 2021). This means that sampling from a Plackett-Luce boils down to sampling $P$ independent Gumbel random variables, which costs $\mathcal{O}(P)$, and then computing an argsort$^K$ of the noised $f(\cdot, x)$ over the discrete action space. We cannot exploit approximate MIPS for this computation as the noise is added after computing the inner product $f(a, x)$, making this step cost $\mathcal{O}(P \log K)$, which makes the total complexity of sampling $\mathcal{O}(P \log K)$, slightly better than the first procedure while compatible with parallel acceleration.

## 2.2 Optimizing The Objective

We want to learn slate policies that can maximize the objective in Equation (4). As our objective is decomposable over contexts, stochastic optimization procedures can be adopted Ruder (2016) making optimization over large datasets possible. For this reason, we can focus on the gradient of the objective for a single context $x$. We derive the score function gradient (Williams, 1992):

$$\nabla_\theta \hat{R}(\pi_\theta|x) = \mathbb{E}_{\pi_\theta(\cdot|x)} [\hat{r}(A_K, x) \nabla_\theta \log \pi_\theta(A_K|x)] \tag{6}$$

$$= \sum_{i=1}^{K} \mathbb{E}_{\pi_\theta(\cdot|x)} [\hat{r}(A_K, x) \nabla_\theta \log \pi_\theta(a_i|x, A_{i-1})]. \tag{7}$$

This gradient is defined as an expectation under $\pi_\theta(\cdot|x)$ over all possible slates. Computing it exactly requires summing $\mathcal{O}(P^K)$ terms which is infeasible. This allows us to approximate the gradient by sampling, which reduces the computation complexity but we will see that this gradient suffers from further problems.

**Computational Burden.** The computational complexity of the gradient is crucial for allowing fast learning of slate as it impacts the running time of every gradient step. Even if we avoid computing the gradient exactly, its approximation can still be a bottleneck when dealing with large action spaces for the following reasons: **(1) Sampling:** Approximating the expectation by sampling slates from $\pi_\theta(\cdot|x)$ can be done in $\mathcal{O}(P \log K)$. However, if the action space is large ($P$ in the order of millions), even a linear complexity on $P$ can be problematic, massively slowing down our optimization procedure. **(2) The Normalizing Constant:** Approximating the gradient needs the computation of $\nabla_\theta \log \pi_\theta(A_K^i|x)$ for the sampled slates $\{A_K^i\}_{i \in [S]}$. This can slow down the optimization procedure as computing the normalizing constant $Z_\theta(x)$ requires summing over all actions, making the complexity of the operation linear in $P$.

We can solve this computational burden by tackling the two previous problems separately. We can get rid of the normalizing constant in the gradient by generalizing the results of Sakhi et al. (2023b). For a single context $x$, we can derive a covariance gradient that does not require $Z_\theta(x)$:

$$\nabla_\theta \hat{R}(\pi_\theta|x) = \mathbb{C}\text{ov}_{A_K \sim \pi_\theta(\cdot|x)} \left[ \hat{r}(A_K, x), \sum_{i=1}^{K} \nabla_\theta f_\theta(a_i, x) \right], \tag{8}$$

with $\mathbb{C}\mathrm{ov}[A, \boldsymbol{B}] = \mathbb{E}[(A - \mathbb{E}[A]).(\boldsymbol{B} - \mathbb{E}[\boldsymbol{B}])]$ a covariance between $A$ a scalar function, and $\boldsymbol{B}$ a vector. The proof of this new gradient expression is developed in Appendix A.1. One can see that for $K = 1$, we recover the results of Sakhi et al. (2023b). This form of gradient does not involve the computation of a normalizing constant, which solves the second problem, but still requires sampling from the policy $\pi_\theta(\cdot|x)$ to get a good covariance estimation. To lower the time complexity of this step, we can use Monte Carlo techniques such as Importance Sampling/Rejection Sampling (Owen, 2013) with carefully chosen proposals to achieve fast sampling without sacrificing the accuracy of the gradient approximation. We develop a discussion around accelerating Plackett-Luce training in Appendix A.1. While we may have ways to deal with the computation complexity, the Plackett-Luce Policy gradient estimate still suffers from the following problems:

**Variance Problems.** Let us focus on the gradient derived in Equation (6). Its exact computation is intractable, and we need to estimate it by sampling from $\pi_\theta(\cdot|x)$. Let us imagine we sample a slate $A_K = [a_1, ..., a_K]$ to estimate the gradient:

$$G_\theta(x) = \hat{r}(A_K, x)\nabla_\theta \log \pi_\theta(A_K|x)$$
$$= \hat{r}(A_K, x)\sum_{i=1}^{K} g_\theta^i(x),$$

with $g_\theta^i(x)$ set to $\nabla_\theta \log \pi_\theta(a_i|x, A_{i-1})$ to simplify the notation. $G_\theta(x)$ is an unbiased estimator of the gradient $\nabla_\theta \hat{R}(\pi_\theta|x)$ and can be used in a stochastic optimization procedure in a principled manner (Ruder, 2016). However, the efficiency of any stochastic gradient descent algorithm depends on the variance of the gradient estimate (Ajalloeian & Stich, 2021), which is defined for vectors $X$ as:

$$\mathbb{V}[X] = \mathbb{E}\left[||X - \mathbb{E}[X]||^2\right] \in \mathbb{R}^+.$$

Gradients with small variances allow practitioners to use bigger step sizes, which reduces the number of iterations as it makes the whole optimization procedure converge faster. Naturally, we would want the variance of our estimator to be small. Unfortunately, the variance of $G_\theta(x)$ grows with the slate size $K$. Writing down the variance w.r.t $\pi_\theta(\cdot|x)$ of $G_\theta(x)$:

$$\mathbb{V}[G_\theta(x)] = \mathbb{V}[\hat{r}(A_K, x)\nabla_\theta \log \pi_\theta(A_K|x)] = \sum_{i=1}^{K} \mathbb{V}[\hat{r}(A_K, x)g_\theta^i(x)] + 2\sum_{i<j} \mathbb{C}\mathrm{ov}[\hat{r}(A_K, x)g_\theta^i(x), \hat{r}(A_K, x)g_\theta^j(x)].$$

The first term of this variance is a sum over the slate of individual variances, which clearly grows as $K$ grows. For the covariance terms, we argue that, especially when initializing the parameter $\theta$ randomly, the gradients $g_\theta^i(x)$ and $g_\theta^j(x)$ will have in expectation different signs making the covariance terms cancel out, leaving the sum of the individual variance terms dominate. This gives a variance that grows in $\mathcal{O}(K)$.

Previous work already showed empirically that the score function gradient for the Plackett-Luce distribution has a large variance (Gadetsky et al., 2020; Buchholz et al., 2022), large enough that learning is not possible in difficult scenarios without considering variance reduction methods (Gadetsky et al., 2020). A possible solution is Randomized Quasi-Monte Carlo (Buchholz et al., 2022; L'Ecuyer, 2016), which produces more accurate estimates by covering the sampling space better. Its value is only significant when we sample few slates $\{A_K^s\}_{s\in[S]}$ to approximate the gradient (Buchholz et al., 2022; L'Ecuyer, 2016). Another direction explores control variates (Gadetsky et al., 2020) as a variance reduction technique. This method requires additional computational costs and a perfect modelling of a differentiable reward proxy to expect variance reduction (Grathwohl et al., 2018).

We want to find a way to both reduce the variance of our method and the computational burden. One of the simplest method to reduce the variance and gain in computation speed is to reduce the number of parameters we want to train (Koch et al., 2021). In our problem of online decision systems, some parameters can be learned independently making policy learning easier (Sakhi et al., 2023b).

### 2.3 Fixing The Action Embeddings

As we are dealing with large action spaces, we are constrained to the following structure on the relevance function $f_\theta$ for fast querying:

$$\forall a \in \mathcal{A} \quad f_\theta(a, x) = h_\Xi(x)^T \beta_a,$$

with both $h_\Xi(x)$ and $\beta_a$ living in an embedding space $\mathbb{R}^L$ with $L \ll P$. The dimension of the matrix $\beta$ is $[L \times P]$, which can be very large when $P$ is large, and dominates $\Xi$ in terms of number of parameters (Koch et al., 2021; Sakhi et al., 2023b). If we can fix the matrix $\beta$, it would benefit our approach both in terms of computational efficiency and variance reduction. Indeed, reducing the number of parameters accelerates learning, makes the problem more identifiable, and reduces drastically the gradient variance as:

$$\mathbb{V}[G_\theta(x)] = \mathbb{E}[||\overline{G}_\theta(x)||^2] = \mathbb{E}[||\overline{G}_\Xi(x)||^2] + \mathbb{E}[||\overline{G}_\beta(x)||^2]$$
$$= \mathbb{V}[G_\Xi(x)] + \mathbb{V}[G_\beta(x)] \gg \mathbb{V}[G_\Xi(x)].$$

with $\overline{G}_\theta(x) = G_\theta(x) - \nabla_\theta \hat{R}(\pi_\theta|x)$ the centred gradient estimate. In many applications (think about information retrieval, recommender systems or ad placement), the action embeddings can be learned from the massive data we have on the actions. In these scenarios, actions boil down to web pages, products we want to recommend or place in an ad. We usually have collaborative filtering signal (Sakhi et al., 2020; Liang et al., 2018) and product descriptions (Vasile et al., 2016) to learn embeddings from. These signals help us obtain good action embeddings $\beta$ and allow us to fix the action matrix before proceeding to the downstream task we are solving. This approach of fixing $\beta$ is not new, Koch et al. (2021) fix $\beta$ to learn a large scale recommender system deployed in production, and Sakhi et al. (2023b) show empirically that learning $\beta$ actually hurts the performance of softmax policies in large scale scenarios.

**We can still learn more about the actions.** Even with $\beta$ fixed, there are sufficient degrees of freedom to solve our downstream task. If we write down:

$$h_\Xi(x) = h_{\Xi'}(x)Z,$$

with $\Xi = [\Xi', Z]$, $Z$ being a learnable matrix of size $[L', L]$. the relevance function can be written as:

$$f_\theta(a, x) = h_{\Xi'}(x)^T(Z\beta_a), \quad \forall a \in \mathcal{A}.$$

This means that, even though the matrix $\beta$ is fixed, we can learn a linear transform of $\beta$ with the help of $Z$, injecting information from the downstream task and learning a transformed representation of the actions in the embedding space. In the rest of the paper, we will fix the action embeddings $\beta$ making the parametrization of the relevance function $f_\theta$ reduce to $\theta = \Xi$, giving for any $x$:

$$f_\theta(a, x) = h_\theta(x)^T \beta_a, \quad \forall a \in \mathcal{A}.$$

## 3 Latent Gaussian Perturbation

Fixing the embeddings helps to decrease the variance and the number of parameters to optimize substantially, which makes our policy learning routine converge faster. This approach, however, does not deal with the fundamental limit of the Plackett-Luce variance, which grows with the slate size $K$. This policy class also needs particular care to accelerate its learning; we should adopt the new gradient formula stated in (8) combined with advanced Monte Carlo techniques to approximate the gradient efficiently, making the implementation of such methods difficult to achieve.

These issues come intrinsically with the adoption of the Plackett-Luce policy, and suggest that we should think differently about how we define policies over slates. Let us investigate this family of policies. For a particular context $x$ and a slate $A_K$, we write down its Gumbel trick (Huijben et al., 2021) expression:

$$\pi_\theta(A_K|x) = \mathbb{E}_{\gamma \sim \mathcal{GP}(0,1)} \left[ \mathbb{1}\left[ A_K = \underset{a' \in \mathcal{A}}{\operatorname{argsort}^K} \left\{ h_\theta(x)^T \beta_{a'} + \gamma_i \right\} \right] \right].$$

The Plackett-Luce policy is a smoothed, differentiable relaxation of the following deterministic policy:

$$b_\theta(A_K|x) = \mathbb{1}\left[A_K = \underset{a'\in\mathcal{A}}{\operatorname{argsort}^K}\left\{h_\theta(x)^T\beta_{a'}\right\}\right]$$
$$= \mathbb{1}\left[A_K = d_\theta(x)\right].$$

$b_\theta$ is a deterministic policy putting all its mass on the actions chosen by our decision function $d_\theta$. Note that, taking an expectation under $b_\theta$ in Equation (4) recovers Equation (3). It means that introducing noise relaxed Equation (3) to a differentiable objective. This relaxation is achieved by randomly perturbing the scores of the different actions with Gumbel noise (Huijben et al., 2021). As this perturbation is done in the action space level, it induces properties that are not desirable: **(1)** The gradient of this policy is an expectation under a potentially large action space, accentuating variance problems. **(2)** The perturbation scales with the size of the action space, as we need $P$ random draws of Gumbel noises. **(3)** Sampling from this policy cannot naturally benefit from approximate MIPS algorithms, as discussed previously.

We observe that the majority of these problems emerge from doing this perturbation in the action space level. With this in mind, we introduce the **LGP: Latent Gaussian Perturbation** policy, that is defined for a context $x$ and a slate $A_K$ by:

$$\pi_\theta^\sigma(A_K|x) = \mathbb{E}_{\epsilon\sim\mathcal{N}(0,\sigma^2 I_L)}\left[\mathbb{1}\left[A_K = \underset{a'\in\mathcal{A}}{\operatorname{argsort}^K}\left\{(h_\theta(x)+\epsilon)^T\beta_{a'}\right\}\right]\right],$$

with $\sigma > 0$, a shared standard deviation across dimensions in the latent space $\mathbb{R}^L$. The **LGP** policy defines a smoothed, differentiable relaxation of the deterministic policy $b_\theta$ by adding Gaussian noise in the latent space $\mathbb{R}^L$. This approach can be generalized by perturbing the latent space with any other continuous distribution $Q$. The resulting method, called **LRP: Latent Random Perturbation** is presented in Appendix A.2. Focusing on **LGP**, this class of policies present desirable properties:

**Fast Sampling.** For a given $x$, sampling from a **LGP** policy boils down to sampling a Gaussian noise and computing an argsort as:

$$A_K \sim \pi_\theta^\sigma(\cdot|x) \iff A_K = \underset{a'\in\mathcal{A}}{\operatorname{argsort}^K}\left\{(h_\theta(x)+\sigma\epsilon_0)^T\beta_{a'}\right\}, \epsilon_0 \sim \mathcal{N}(0,I_L).$$

As the action embeddings $\beta$ are fixed, and we are performing a perturbation in the latent space, sampling can be done by calling approximate MIPS on the perturbed query $h_\theta^\epsilon(x) = h_\theta(x) + \epsilon$, achieving a complexity of $\mathcal{O}(\log P)$ and improving on the sampling complexity $\mathcal{O}(P\log K)$ of the Plackett-Luce family.

**Well behaved gradient.** Similar to the gradient under the Plackett-Luce policy (6), we can derive a score function gradient for **LGP** policies. Let $A_K\{h\} = \operatorname{argsort}^K_{a'\in\mathcal{A}}\left\{h^T\beta_{a'}\right\}$ the MIPS result for query $h$. Let us write the expected reward under $\pi_\theta^\sigma$ for a particular context $x$:

$$\hat{R}(\pi_\theta^\sigma|x) = \mathbb{E}_{A_K\sim\pi_\theta^\sigma(\cdot|x)}\left[\hat{r}(A_K,x)\right]$$
$$= \mathbb{E}_{\epsilon\sim\mathcal{N}(0,\sigma^2 I_L)}\left[\hat{r}(A_K\{h_\theta(x)+\epsilon\},x)\right]$$
$$= \mathbb{E}_{h\sim\mathcal{N}(h_\theta(x),\sigma^2 I_L)}[\hat{r}(A_K\{h\},x)].$$

The last equality allows us to derive the following gradient:

$$\nabla_\theta\hat{R}(\pi_\theta^\sigma|x) = \nabla_\theta\mathbb{E}_{h\sim\mathcal{N}(h_\theta(x),\sigma^2 I_L)}[\hat{r}(A_K\{h\},x)]$$
$$= \mathbb{E}_{h\sim\mathcal{N}(h_\theta(x),\sigma^2 I_L)}[\hat{r}(A_K\{h\},x)\nabla_\theta\log q_\theta(x,h)],$$

with $\log q_\theta(x,h)$ the log density of $\mathcal{N}(h_\theta(x),\sigma^2 I_L)$ evaluated in $h$. We can obtain an **unbiased** gradient estimate by sampling $\epsilon_0\sim\mathcal{N}(0,I_L)$, setting $h = h_\theta(x)+\sigma\epsilon_0$ and computing:

$$G_\theta^\sigma(x) = \hat{r}(A_K\{h\},x)\nabla_\theta\log q_\theta(x,h)$$
$$= \frac{1}{\sigma}\hat{r}(A_K\{h\},x)\nabla_\theta(\epsilon_0^T h_\theta(x)) \tag{9}$$

This gradient expression solves all issues that the Plackett-Luce gradient estimate suffered from:

- **Fast Gradient Estimate.** This gradient can be approximated in a sublinear complexity $\mathcal{O}(\log P)$. Building an estimator of the gradient follows these three steps: **(1)** We sample $\epsilon_0 \sim \mathcal{N}(0, I_L)$, which is done in a complexity $\mathcal{O}(L) \ll \mathcal{O}(P)$. **(2)** We evaluate the gradient of $\epsilon_0^T h_\theta(x)$ in $\mathcal{O}(L)$ with no dependance on $P$. **(3)** We generate the slate $A_K\{h\}$. This boils down to computing an argsort that can be accelerated using approximate MIPS technology, giving a complexity of $\mathcal{O}(\log P)$.

- **Better Variance. LGP**'s gradient estimate have better statistical properties for two main reasons: **(1)** The gradient is defined as an expectation under a continuous distribution on the latent space $\mathbb{R}^L$, instead of an expectation under a large discrete action space $\mathcal{A}$ of size $P \gg L$. This can have an impact on the variance of the gradient estimator as the sampling space is smaller. **(2)** The approximate gradient defined in Equation (9) does not depend on the slate size $K$. This results in a variance that does not grow with $K$, which will translate to substantial performance gains on problems with larger slates. However, the expression of this gradient suggests that our attention should be directed towards the standard deviation $\sigma$ instead, as the variance of the gradient estimate defined in Equation (9) will scale in $\mathcal{O}(1/\sigma^2)$.

Even if $\sigma$ can be treated as an additional parameter, we fix it to $\sigma = 1/L$ in all our experiments for a fair comparison. The resulting approach will be hyper-parameter free, and will show both statistical and computational benefits. We give a sketch of the resulting optimization procedure in Algorithm 1. This procedure is easy to implement in any automatic differentiation package (Paszke et al., 2019) and is compatible with stochastic first order optimization algorithms (Ruder, 2016). In the next section, we will measure the benefits of the proposed algorithm in different scenarios.

---

**Algorithm 1:** Learning with Latent Gaussian Perturbation

**Inputs:** $D = \{x_i\}_{i=1}^N$, reward estimator $\hat{r}$, the action embeddings $\beta$
**Parameters:** $T \geq 1$, Monte Carlo samples number $S \geq 1$
**Initialise:** $\theta = \theta_0$, MIPS index of $\beta$, $\sigma = 1/L$
**for** $t = 0$ **to** $T - 1$ **do**
    sample a context $x \sim D$
    sample $S$ standard Gaussian noises $\epsilon_1, ..., \epsilon_S \sim \mathcal{N}(0, I_L)$
    compute for $s \in [S]$, $h_s = h_\theta(x) + \epsilon_s$
    compute slates $A_K\{h_s\}$ for $s \in [S]$ with MIPS
    **Estimate the gradient:**

$$grad_\theta \leftarrow \frac{1}{S\sigma} \sum_{s=1}^S \hat{r}(A_K\{h_s\}, x)\nabla_\theta(\epsilon_s^T h_\theta(x))$$

    **Update the policy parameter** $\theta$:
    $\theta \leftarrow \theta - \alpha grad_\theta$
**end**
**return** $\theta$

---

## 4 Experiments

### 4.1 Experimental Setting

For our experiments, we focus on learning slate decision functions for the particular case of recommendation as collaborative filtering datasets are easily accessible, facilitating the reproducibility of our results. We choose three collaborative filtering datasets with varying action space size, MovieLens25M (Harper & Konstan, 2015), Twitch (Rappaz et al., 2021) and GoodReads (Wan & McAuley, 2018; Wan et al., 2019). We process these datasets to transform them into user-item interactions of shape $[U, P]$ with $U$ and $P$ the number of users and actions. The statistics of these datasets are described in Table 2.

We follow the same procedure as Sakhi et al. (2023b) to build our experimental setup. Given a dataset, we split randomly the user-item interaction session $[X, Y]$ into two parts; the observed interactions $X$ and the hidden interactions $Y$. the observed part $X$ represents all the information we know about the user, and will be used by our policy $\pi_\theta$ to deliver slates of interest. The hidden part $Y$ is used to define a reward function that will drive the policy to solve a form of session completion task. For a given slate $A_K = [a_1, ..., a_K]$, we define the reward as:

$$\hat{r}(A_K, X) = \sum_{k=1}^{K} \frac{\mathbb{1}[a_k \in Y]}{2^{k-1}}.$$

Although we can adopt an arbitrary form for the reward function, we want rewards that depend on the whole slate (Aouali et al., 2023b) and that take into account the ordering of the items. We choose a linear reward to be able to compare our method to `PL-Rank` (Oosterhuis, 2022), which exploits the reward structure to improve the training of Plackett-Luce policies. The objective we want to optimize is the following:

$$\hat{R}(\pi_\theta) = \frac{1}{U} \sum_{i=1}^{U} \mathbb{E}_{A_K \sim \pi_\theta(\cdot|X_i)} \left[ \hat{r}(A_K, X_i) \right].$$

The next step is to parametrize the policy $\pi_\theta$. For large scale problems, and for a given $X$, we are restricted to use the following parametrization of the relevance function $f_\theta$:

$$f_\theta(a, X) = h_\theta(X)^T \beta_a, \quad \forall a \in \mathcal{A}.$$

Given the observed interactions $X$, we compute the action embeddings $\beta$ using an SVD matrix decomposition (Klema & Laub, 1980). This allows us to project the different action into a latent space of lower dimension $L \ll P$, making $\beta$ of dimension $[L, P]$. In all experiments, $\beta$ will be fixed unless we want to study the impact of training the embeddings. When $\beta$ is fixed, we create an approximate MIPS index using the HNSW algorithm (Malkov & Yashunin, 2020) with the help of the FAISS library (Johnson et al., 2019). This index will accelerate decision-making online as described in Equation (2) and can also be exploited to speed up the training of **LGP** policies. With $\beta$ defined, we still need to parametrize the user embedding function $h_\theta$. Given $X$, we first define the mean embedding function $M : \mathcal{X} \rightarrow \mathbb{R}^L$:

$$M(X) = \frac{1}{|X|} \sum_{a \in X} \beta_a.$$

The function $M$ computes the average of the item embeddings the user interacted with in $X$ (Koch et al., 2021). $h_\theta$ follows as:

$$h_\theta(X) = M(X)\theta \tag{10}$$

with $\theta$ a parameter of dimension $[L, L]$, much smaller than $[L, P]$, the dimension of $\beta$. All policies in these experiments will use this parametrization. Experiments with deep policies can be found in Appendix A.3.4. The training is conducted on a CPU machine, using the Adam optimizer (Kingma & Ba, 2014), with a batch size of 32. We tune the learning rate on a validation set for all algorithms. We adopt Algorithm 1 to train **LGP** and its accelerated variant **LGP-MIPS**. We denote by **PL-PG**, the algorithm that trains the Plackett-Luce policy trained with the score function gradient; we sample $S \geq 1$ slates $\{A_K^1, ..., A_K^S\}$ from $\pi_\theta$ to derive the gradient estimate for a given $X$:

$$G_\theta^S(X) = \frac{1}{S} \sum_{i=1}^{S} \hat{r}(A_K^s, X) \nabla_\theta \log \pi_\theta(A_K^s | X). \tag{11}$$

We also compare our results to `PL-Rank` (Oosterhuis, 2022) that exploits the linearity of the reward to have a better gradient estimate. As we are mostly interested in the performance of the decision system $d_\theta$, all the rewards reported in the experiments are computed using:

$$\hat{R}(d_\theta) = \frac{1}{U} \sum_{i=1}^{U} \hat{r}(d_\theta(X_i), X_i).$$

|  | #Actions | #Users | Interactions Density |
|---|---|---|---|
| **MovieLens 25M** | 55K | 162K | 0.24% |
| **Twitch** | 750K | 580K | 0.008% |
| **GoodReads** | 2.23M | 400K | 0.01% |

**Table 2:** The statistics of the datasets after preprocessing

In the next section, we study empirically the performance of these approaches by training them with the same time budget on the different datasets. Additional experiments can be found in Appendix A.3 to confirm the robustness of our results and better understand the behaviour of both the Plackett-Luce policy and the newly introduced **LGP** policy.

### 4.2 Performance under the same time budget

To measure the performance of our algorithms, we use all three datasets with their statistics described in Table 2. We fix the latent space dimension $L = 100$ and use a slate size of $K = 5$ for these experiments. We split each dataset by users and keep 10% to create a validation set, on which the reward of the decision function is reported. As these algorithms present different iteration speeds, we fix the same time budget for all training methods for a fair comparison. Training with a time budget also simulates a real production environment, where practitioners are bounded by time constraints and scheduled deployments. For all datasets and training routines, we allow a runtime of 60 minutes, and evaluate our policies on the validation set for 10 equally spaced intervals. The results of these experiments are presented in Figure 1 where we plot the evolution of the validation reward on all datasets, for different values of $S \in \{1, 10, 100\}$; the number of samples used to approximate the gradient.

Our first observation from the graph is that `PL-PG` cannot compete with other algorithms, even in the simplest scenario of the MovieLens dataset. Its poor performance is mainly due to the high variance of its gradient estimate. This is confirmed by the performance of `PL-Rank`. Indeed, the `PL-Rank` algorithm works with the same policy class, has the same iteration cost (scales linearly in $P$) and only differs on the quality of the gradient estimate; exploiting the structure of the reward allow us to obtain an estimate with lower variance. These results confirm our first intuition. `PL-PG` suffer from large variance problems (even in modest sized problems) and is not suitable to solve large scale slate decision problems.

Let us now focus on our newly proposed algorithms; `LGP` and its accelerated variant `LGP-MIPS`. We observe that in all scenarios considered, the acceleration brought by the approximate MIPS index benefits our algorithm in terms of performance. For the same time budget, `LGP-MIPS` obtains better reward than `LGP`, with the biggest differences observed on datasets with large action spaces; Twitch and GoodReads. `LGP-MIPS` always gives the best performing decision function, for all datasets and number of Monte Carlo samples $S$ considered. These results are promising as our algorithm which is agnostic to the form of the reward outperforms `PL-Rank` that is solely designed to tackle the particular case of linear rewards. This performance is due to `LGP-MIPS`'s superior sampling complexity combined with an unbiased, low variance gradient estimate. It is also worthy to note that we were unable to run algorithms optimizing Plackett-Luce policies (`PL-PG` and `PL-Rank`) on the GoodReads dataset with $S = 100$ due to its massive memory footprint. As sampling is done on the action space, **Plackett-Luce**-based methods need for each iteration samples of size $\mathcal{O}(SP)$ compared to **LGP**-based method for which the sampling is done in the latent space requiring $\mathcal{O}(SL)$ memory usage. Being able to increase the number of Monte Carlo Samples $S$ is desirable, as it helps reduce the variance of the gradient estimates and accelerates further the training.

These results demonstrate the utility of our newly proposed method over Plackett-Luce for learning slate decision systems. The **LGP** policy class combined with accelerated MIPS indices produces unbiased, low variance gradient estimates that are fast to compute, can scale to massive action spaces and exhibit low memory usage, making our algorithm the best candidate for optimizing large scale slate decision systems.

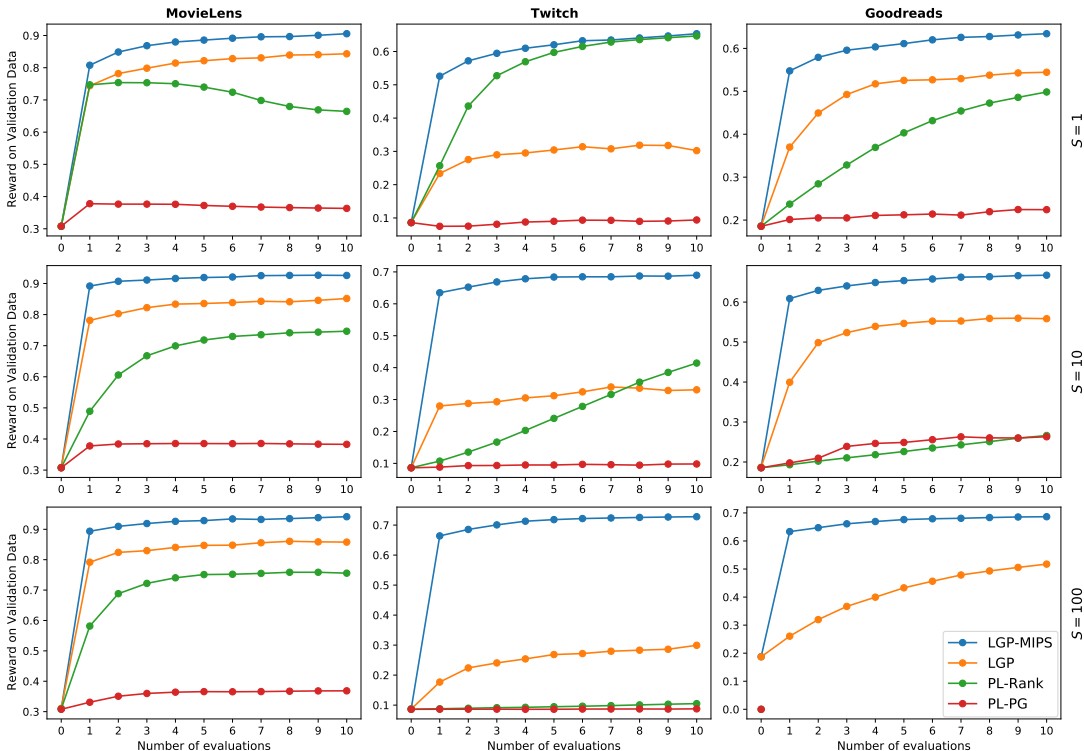

**Figure 1:** The performance of slate decision functions obtained after optimizing them by different training algorithms for the same time budget of 60 minutes. Each evaluation on the validation data is done after 6 minutes of training.

### 4.3 Performance under the same number of iterations

We proposed a new family of policies that enables fast optimization, and naturally benefits from unbiased gradient estimates with a variance that does not depend on the slate size $K$. The results reported in Figure 1 show that the newly proposed methods are suitable for optimizing large scale decision systems, under strict time constraints. We provide further experiments and compare the performance of the obtained policies after training them, with different approaches, under the same number of iterations. This gives insight into the behaviour of the variance of the gradient approximations while neglecting the iteration cost. For the same learning rate, gradient approximations with low variance need less optimization iterations to converge. As `PL` methods tend to iterate slowly, especially in large action space scenarios (Twitch and Goodreads), we fix the number of iterations in all experiments to the iterations made by `PL-PG` after 60 minutes of optimization. We report the evolution of the training reward of all methods, for different settings, in Figure 2. We identify two evolution patterns from the plots. A first, fast convergence pattern that `PL-Rank` alongside the `LGP` family enjoy, and a second, much slower convergence pattern of `PL-PG`. This difference in training reward evolution conveys the following message; the `LGP` family benefit from a gradient estimate with a variance similar to `PL-Rank`, and much lower than the variance of the `PL-PG` gradient estimate. In addition, not only can the `LGP` family iterates faster, but it enjoys low variance gradient estimates and does not rely on a linearity assumption on the reward. This further confirms the improvements brought by the `LGP` family, motivating it as an excellent alternative to Plackett-Luce for optimize large scale decision systems.

## 5 Related work

**Learning from interactions.** Recent advances in learning large scale decision systems adopt the offline Contextual Bandit/Reinforcement Learning framework (Chen et al., 2022; Wang et al., 2022; Ma et al., 2020; Chen et al., 2019) proving itself as a powerful paradigm to align business metrics with the offline optimization problem. Research in this direction either explore the use of known policy learning algorithms (Chen et al.,

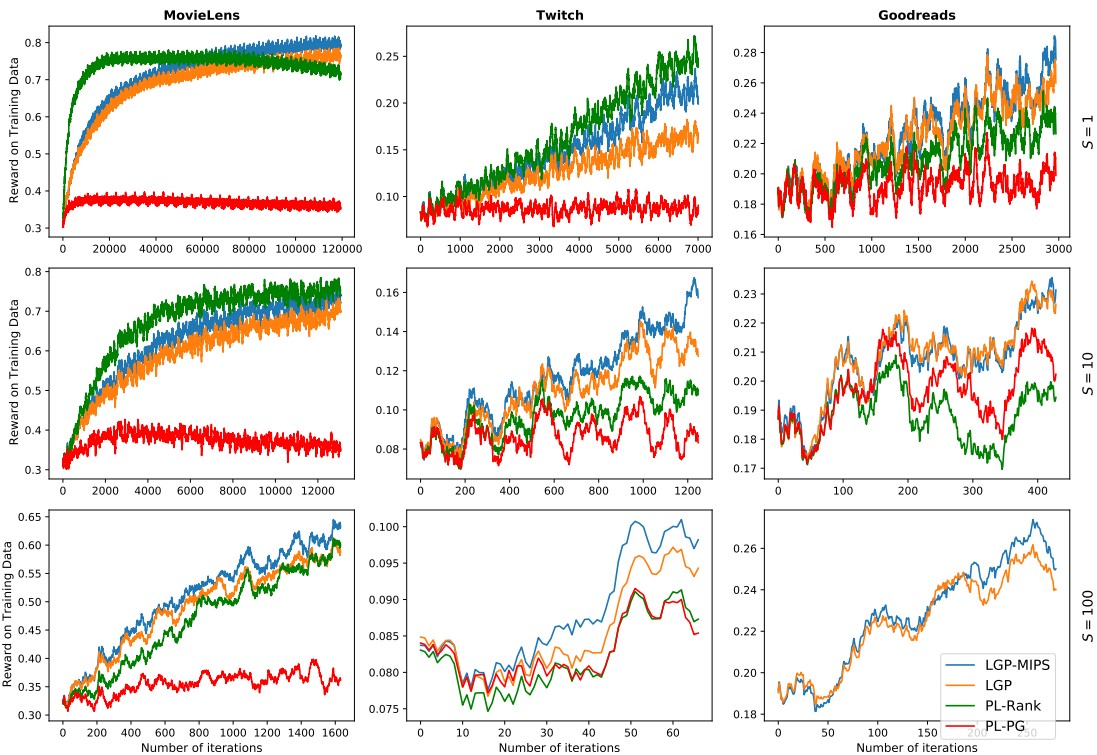

**Figure 2:** Comparing the performance of slate decision functions obtained by running our different training algorithms for the same iterations. The newly proposed relaxation `LGP` have a gradient estimate of quality comparable to `PL-Rank` as they result in policies with a similar training behaviour. `LGP` methods enjoy low variance gradient estimate without making an assumption on the structure of the reward, contrary to `PL-Rank`.

2022) for learning online decision systems or define better rewards to guide this learning (Wang et al., 2022; Christakopoulou et al., 2022). Chen et al. (2019) showed that large scale recommender systems can benefit from the contextual bandit formulation and introduced a correction to the REINFORCE gradient to encourage softmax policies to recommend multiple items. Their method can be seen as a heuristic applicable when the slate level reward is decomposable into a sum of single item rewards. This assumption is violated in settings where strong interactions exist between the items in the slate. Our method is versatile, does not assume any structure on the reward, and is able to optimize slate decision systems by introducing a new relaxation, smoothing them into a policy that has a better learning behavior than classical Plackett-Luce.

**Scalability.** The question of scaling offline policy learning to large scale decision systems has received limited attention. It has been shown that offline softmax policy learning can be scaled to production systems (Chen et al., 2019). Recently, Sakhi et al. (2023b) focused on studying the scalability of optimizing policies tailored to one item recommendation, using the covariance gradient estimator combined with importance sampling (Owen, 2013) to exploit approximate MIPS technology in the training phase. Their gradient approximation is provably biased, sacrificing the convergence guarantees provided by stochastic gradient descent (Ruder, 2016). Our method extends the scope of this paper, as it can be applied to slate recommendation. It provides a simpler relaxation than Plackett-Luce, producing a learning algorithm that benefits from approximate MIPS technology, naturally obtaining unbiased gradient estimates with better statistical properties.

**Learning to Rank.** The learning to rank literature separates algorithms by the output space they operate on, making a clear distinction between pointwise, pairwise and listwise approaches (Liu, 2009). Our method falls in the latter (Xia et al., 2008), as we operate with policies on the slate level. The majority of work in LTR trains decision systems through the optimization of ranking losses (Wang et al., 2018; Oosterhuis, 2022), defined as differentiable surrogates of ranking metrics or by an adapted *maximum likelihood estimation* (Rendle et al., 2009; Ma et al., 2021). In the same direction, differentiable sorting algorithms (Grover et al.,

2019; Prillo & Eisenschlos, 2020) aim at producing a differentiable relaxation to sorting functions that handle reward on the item level These methods also require linear rewards in addition to training an item-item interaction matrix, *quadratic* on the action space size, making them unsuitable to massive action spaces. We are interested in training reward-driven, large scale slate decision systems, to learn rankings that are more aligned with arbitrary, complex rewards functions.

**Smoothing non-differentiable objectives.** Our procedure can be interpreted as a stochastic relaxation of non-differentiable decision functions. This relaxation is achieved by introducing a well chosen noise in the latent space. PAC-Bayesian policy learning (London & Sandler, 2019; Sakhi et al., 2023a; Aouali et al., 2023a) and Black-Box optimization algorithms (Bajaj et al., 2021; Rechenberg, 1978; Staines & Barber, 2012) adopt a similar paradigm to optimize non-smooth loss functions. They proceed by injecting noise in the parameter space and derive a gradient with respect to the noise distribution. This parameter level perturbation can suffer from computational issues when the number of parameters increases. Our method is agnostic to the number of parameters. By perturbing the latent space directly, we bypass this problem and simplify the sampling procedure resulting in faster and more efficient training.

## 6 Conclusion and future work

Countless large scale online decision systems are tasked with delivering *slates* based on contextual information. We formulate the learning of these systems in the offline contextual bandit setting. This framework relaxes decision systems to stochastic policies, and proceeds at learning them through REINFORCE-like algorithms. Plackett-Luce provides an interpretable distribution over ordered lists of items but its use in a large scale policy learning context is far from being optimal. In this paper, we motivate the **Latent Gaussian Perturbation**, a new policy class over ordered lists defined as a stochastic, differentiable relaxation of the argsort decision function, induced by perturbing the latent space. The **LGP** policy provides gradient estimates with better computational and statistical properties. We built an intuition on why this new policy class is better behaved and demonstrated through extensive experiments that not only **LGP** is faster to train, considerably reducing the computational cost, but it also produces policies with better performance making the use of Plackett-Luce policies in this context obsolete. This work gives practitioners a new way to address large scale slate policy learning and aim at contributing into the adoption of REINFORCE decision systems. The results obtained in this paper suggest that the relaxations used to define policies have a big role in optimization and that we might need to take a step backward and reconsider the massively adopted Softmax/Plackett-Luce parametrization. As we only focused on simple noise distributions, a nice avenue of research will be to study the impact of the choice of these distributions on the learning algorithm produced.

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

# A Appendix

## A.1 Accelerating Plackett-Luce training

---

**Algorithm 2:** Categorical Distribution: Rejection sampling using MIPS

---

**Input:** $h_\Xi$, $\beta$, $x$, and indexes on $\beta$ and parameter $K$, catalogue size $\mathcal{P}$

**Output:** $a$ which is a sample from $P(A = a) = \frac{\exp(h_\Xi(x)^T \beta_a)}{\sum_{a'} \exp(h_\Xi(x)^T \beta_{a'})}$

$\alpha_1, ..., \alpha_K = \mathrm{argsort}(h_\Xi(x)^T \beta)_{1:K}$

$\mathcal{Z}' = \mathcal{P} \exp(h_\Xi(x)^T \beta_{\alpha_K})$

$\mathcal{Z}'' = \sum_{a'}^K \exp(h_\Xi(x)^T \beta_{a'}) - \exp(h_\Xi(x)^T \beta_{\alpha_K})$

$P_K = [\exp(h_\Xi(x)^T \beta_{\alpha_1})/\mathcal{Z}'', ..., \exp(h_\Xi(x)^T \beta_{\alpha_K})/\mathcal{Z}'']$

**while** *True* **do**

    $d \sim \mathrm{cat}([\mathrm{tail}, \mathrm{head}], [\frac{\mathcal{Z}'}{\mathcal{Z}'+\mathcal{Z}''}, \frac{\mathcal{Z}''}{\mathcal{Z}'+\mathcal{Z}''}])$

    **if** *d=head* **then**

        $r \sim \mathrm{cat}(\alpha_1, ..., \alpha_K, P_K)$

        **return** $r$

    **end**

    **if** *d=tail* **then**

        Sample $q$ uniformly from the set $\{1, ..., P\}$

        Sample $u$ from a uniform distribution

        **if** $\frac{\exp(h_\Xi(x)^T \beta_q)}{\exp(h_\Xi(x)^T \beta_{\alpha_K})} > u$ **then**

            **return** $q$

        **end**

    **end**

**end**

---

As it was discussed in the main paper, we can derive a covariance gradient that does not require the computation of the normalizing constant $Z_\theta(x)$:

$$\nabla_\theta \hat{R}(\pi_\theta|x) = \mathbb{Cov}_{A_K \sim \pi_\theta(.|x)} \left[ \hat{r}(A_K, x), \sum_{i=1}^K \nabla_\theta f_\theta(a_i, x) \right],$$

with $\mathbb{Cov}[A, \boldsymbol{B}] = \mathbb{E}[(A - \mathbb{E}[A]).(\boldsymbol{B} - \mathbb{E}[\boldsymbol{B}])]$ a covariance between $A$ a scalar function, and $\boldsymbol{B}$ a vector. The proof follows:

$$\nabla_\theta \hat{R}(\pi_\theta|x) = \mathbb{E}_{\pi_\theta(\cdot|x)}[\hat{r}(A_K, x)\nabla_\theta \log \pi_\theta(A_K|x)]$$

$$= \sum_{i=1}^K \mathbb{E}_{\pi_\theta(\cdot|x)}[\hat{r}(A_K, x)\nabla_\theta \log \pi_\theta(a_i|x, A_{i-1})]$$

$$= \sum_{i=1}^K \mathbb{E}_{\pi_\theta(\cdot|x)} \left[ \hat{r}(A_K, x) \left( \nabla_\theta f_\theta(a_i, x) - \nabla_\theta \log Z_\theta^{i-1}(x) \right) \right]$$

Using the log trick, we derive the following equality:

$$\nabla_\theta \log Z_\theta^{i-1}(x) = \mathbb{E}_{\pi_\theta(a_i|x, A_{i-1})}[\nabla_\theta f_\theta(a, x)].$$

This equality is then injected in the gradient formula derived above to obtain:

$$\nabla_\theta \hat{R}(\pi_\theta|x) = \mathbb{Cov}_{A_K \sim \pi_\theta(.|x)} \left[ \hat{r}(A_K, x), \sum_{i=1}^K \nabla_\theta f_\theta(a_i, x) \right].$$

This concludes the proof. One can see that for $K = 1$, we recover the results of Sakhi et al. (2023b). This form of gradient still requires sampling from the policy $\pi_\theta(\cdot|x)$ to get a good covariance estimation. To lower the time complexity of this step, we can use Monte Carlo techniques such as Importance Sampling/Rejection Sampling (Owen, 2013) with carefully chosen proposals to achieve fast sampling without sacrificing the accuracy of the gradient approximation.

In Sakhi et al. (2023b), a softmax policy learning algorithm was accelerated by approximating the gradients using a self normalized importance sampling algorithm with a proposal distribution that can both exploit the MIPS structure and is a good approximation of the target softmax distribution. This idea can also be used to motivate a rejection sampling algorithm, as a similar proposal can be shown to form an envelope of the target density. It can also be extended from softmax to Plackett-Luce. When the idea of rejection sampling is combined with the MIPS proposal, it results in the rejection sampling algorithm shown in Algorithm 2. While Algorithm 2 can be extended to the slate policy case, enabling the fast evaluation of the covariance gradient estimator, it will still suffer from high variance gradient estimates.

## A.2 LRP: Latent Random Perturbation

The method presented in the main paper can be generalized. Instead of focusing on Gaussian distributions, the latent perturbation can be done by any continuous distribution resulting in the more general **LRP: Latent Random Perturbation** policy. For a context $x$ and a slate $A_K$, its expression is given by:

$$\pi_\theta^\mathcal{Q}(A_K|x) = \mathbb{E}_{\epsilon \sim \mathcal{Q}} \left[ \mathbb{1} \left[ A_K = \operatorname*{argsort}_{a' \in \mathcal{A}}^{\mathrm{K}} \left\{ (h_\theta(x) + \epsilon)^T \beta_{a'} \right\} \right] \right],$$

with $\mathcal{Q}$ a continuous distribution on the latent space $\mathbb{R}^L$. the **LRP** policy defines a smoothed, differentiable relaxation of the deterministic policy $b_\theta$ by adding noise in the latent space $\mathbb{R}^L$. This class of policies present desirable properties:

**Fast Sampling.** For a given $x$, sampling from a **LRP** policy boils down to sampling from $\mathcal{Q}$ and computing an argsort as:

$$A_K \sim \pi_\theta^\mathcal{Q}(\cdot|x) \iff A_K = \operatorname*{argsort}_{a' \in \mathcal{A}}^{\mathrm{K}} \left\{ (h_\theta(x) + \epsilon)^T \beta_{a'} \right\}, \epsilon \sim \mathcal{Q}.$$

Let us suppose the sampling from $\mathcal{Q}$ is easy. As the action embeddings $\beta$ are fixed, and we are performing a perturbation in the latent space, we can set $h_\theta^\epsilon(x) = h_\theta(x) + \epsilon$ which makes sampling compatible with approximate MIPS technology making the sampling achievable in $\mathcal{O}(\log P)$, better than the sampling complexity $\mathcal{O}(P \log K)$ of the Placett-Luce family.

**Well behaved gradient.** Similar to the gradient under the Plackett-Luce policy (6), we can derive a score function gradient for **LRP** policies. For a given $x$, let us write its expected reward under $\pi_\theta^\mathcal{Q}$:

$$
\begin{aligned}
\hat{R}(\pi_\theta^\mathcal{Q}|x) &= \mathbb{E}_{A_K \sim \pi_\theta^\mathcal{Q}(\cdot|x)} \left[ \hat{r}(A_K, x) \right] \\
&= \mathbb{E}_{\epsilon \sim \mathcal{Q}} \left[ \hat{r}(A_K\{h_\theta(x) + \epsilon\}, x) \right] \\
&= \mathbb{E}_{h \sim \mathcal{Q}_\theta(x)} [\hat{r}(A_K\{h\}, x)]
\end{aligned}
\qquad
\begin{aligned}
h \sim \mathcal{Q}_\theta(x) &\iff h = h_\theta(x) + \epsilon, \quad \epsilon \sim \mathcal{Q} \\
A_K\{h\} &= \operatorname*{argsort}_{a' \in \mathcal{A}}^{\mathrm{K}} \left\{ h^T \beta_{a'} \right\}.
\end{aligned}
$$

$\mathcal{Q}_\theta(x)$ is the induced distribution on the user embeddings. $\mathcal{Q}_\theta(x)$ has a tractable density if we choose a classical noise distribution $\mathcal{Q}$ (e.g. Gaussian). Working with the induced distribution $\mathcal{Q}_\theta(x)$ transforms the learning parameters $\theta$ to parameters of the distribution, allowing us to derive the following gradient:

$$
\begin{aligned}
\nabla_\theta \hat{R}(\pi_\theta^\mathcal{Q}|x) &= \nabla_\theta \mathbb{E}_{h \sim \mathcal{Q}_\theta(x)} [\hat{r}(A_K\{h\}, x)] \\
&= \mathbb{E}_{h \sim \mathcal{Q}_\theta(x)} [\hat{r}(A_K\{h\}, x) \nabla_\theta \log q_\theta(x, h)],
\end{aligned}
$$

with $\log q_\theta(x, h)$ the log density of $\mathcal{Q}_\theta(x)$ evaluated in $h$. We can obtain an unbiased gradient estimate by sampling $h \sim \mathcal{Q}_\theta(x)$:

$$G_\theta^{\mathcal{Q}}(x) = \hat{r}(A_K\{h\}, x)\nabla_\theta \log q_\theta(x, h). \tag{12}$$

This gradient expression solves all issues that the Plackett-Luce gradient estimate suffered from:

- **Fast Gradient Estimate.** This gradient can be approximated in a sublinear complexity $\mathcal{O}(\log P)$. Building an estimator of the gradient follows these three steps: **(1)** We sample $h \sim \mathcal{Q}_\theta(x)$, which boils down to adding the noise $\epsilon \sim \mathcal{Q}$ to $h_\theta(x)$. This can be done in a complexity $\mathcal{O}(L) \ll \mathcal{O}(P)$ if $\mathcal{Q}$ is chosen properly. **(2)** We evaluate the gradient of the log density $\nabla_\theta \log q_\theta(x, h)$ on $h$. With a well-chosen $\mathcal{Q}$, this can be done in $\mathcal{O}(L)$ as we do not need to normalize over a large discrete action space. **(3)** We generate the slate $A_K\{h\}$. This boils down to computing an argsort which can be accelerated using approximate MIPS technology, giving a complexity of $\mathcal{O}(\log P)$.

- **Better Variance. LRP**'s gradient estimate have better statistical properties for two main reasons: **(1)** The gradient is defined as an expectation under a continuous distribution on the latent space $\mathbb{R}^L$, instead of an expectation under a large discrete action space $\mathcal{A}$ of size $P \gg L$. This can have an impact on the variance of the gradient estimator as the sampling space is smaller. **(2)** The approximate gradient defined in Equation (12) does not depend on the slate size $K$. This results in a variance that does not grow with $K$. This means that we will notice substantial gains when training policies with larger slates.

### A.3 Additional Experiments

### A.3.1 Robustness of the results

We conduct a further simulation study to be sure that the results obtained in the main paper are robust both to initialisation and to the slate size $K$. To this end, we repeat the same experimental setup as in the performance subsection; we use all three datasets, and we fix the latent space dimension $L = 100$. We run the optimization of our algorithms for 10 minutes and evaluate the obtained policy on the validation split. For each dataset, we define two settings; a setting with a moderate slate size of $K = 5$ and one with a bigger slate size of $K = 100$. The training procedure is repeated for 6 different seeds, and we present the average performance and the standard error for different values of $S \in \{1, 10, 100\}$; the number of samples used to approximate the gradient. We present results of two different natures:

- The performance on the validation set, of the obtained policies after training them with different algorithms, under the same time constraint of 10 minutes, is reported in Table 3.

- The performance on the training set, of the obtained policies, after training them for the same number of iterations. As training with `PL` algorithms can be very slow, we fix the number of iterations to the iterations done by `PL-PG` in 10 minutes. The results of these experiments are reported in Table 4.

Let us first focus on the performance obtained after training policies under the same time budget. Looking at Table 3, we can clearly observe that `LGP`, especially its `MIPS` accelerated version, always yields the best performing policies no matter the setting adopted. The obtained results are significant, as the standard error is small compared to the difference of performance between the methods. These observations confirm that the `LGP` family is perfectly suited for industrial applications, as it iterates faster and is robust to different settings. It is noteworthy that the `PL` family cannot run with big Monte Carlo samples $S$, especially in applications with large action spaces, as it suffers from a large memory footprint.

For the sake of completeness, we also report in Table 4, the performance of our decision systems after training them with different approaches under a fixed number of iterations. One can observe that `PL-Rank` results in the best decision systems, followed by the `LGP` family. These results suggest that by exploiting the linearity of the reward, the gradient estimates derived by `PL-Rank` enjoy a better variance than the gradient estimates

of the `LGP` family. The application of `PL-Rank` however is only restricted to linear rewards, contrary to our newly proposed approach that does not make any assumptions on the structure of the reward. This means that when the reward is complex, `LGP` methods should be adopted, as they offer a better alternative to the naive, gradient estimate of `PL-PG`.

| Algorithm | $S$ | MovieLens | | Twitch | | GoodReads | |
|---|---|---|---|---|---|---|---|
| | | $K = 5$ | $K = 100$ | $K = 5$ | $K = 100$ | $K = 5$ | $K = 100$ |
| LGP-MIPS | $S = 1$ | $0.834 \pm 0.003$ | $0.599 \pm 0.002$ | $0.438 \pm 0.009$ | $0.343 \pm 0.005$ | $0447 \pm 0.002$ | $0.396 \pm 0.007$ |
| | $S = 10$ | $\mathbf{0.866 \pm 0.003}$ | $\mathbf{0.881 \pm 0.003}$ | $0.567 \pm 0.007$ | $0.563 \pm 0.004$ | $0.539 \pm 0.004$ | $0.554 \pm 0.004$ |
| | $S = 100$ | $\mathbf{0.865 \pm 0.002}$ | $0.867 \pm 0.002$ | $\mathbf{0.607 \pm 0.005}$ | $0.609 \pm 0.005$ | $\mathbf{0.580 \pm 0.001}$ | $\mathbf{0.592 \pm 0.002}$ |
| LGP | $S = 1$ | $0.753 \pm 0.005$ | $0.776 \pm 0.004$ | $0.171 \pm 0.004$ | $0.195 \pm 0.004$ | $0.259 \pm 0.013$ | $0.278 \pm 0.013$ |
| | $S = 10$ | $0.716 \pm 0.004$ | $0.744 \pm 0.005$ | $0.133 \pm 0.002$ | $0.152 \pm 0.003$ | $0.227 \pm 0.012$ | $0.244 \pm 0.012$ |
| | $S = 100$ | $0.700 \pm 0.003$ | $0.721 \pm 0.003$ | $0.097 \pm 0.002$ | $0.111 \pm 0.003$ | $0.192 \pm 0.010$ | $0.204 \pm 0.010$ |
| PL-Rank | $S = 1$ | $0.750 \pm 0.004$ | $0.769 \pm 0.005$ | $0.103 \pm 0.003$ | $0.113 \pm 0.003$ | $0.188 \pm 0.009$ | $0.200 \pm 0.010$ |
| | $S = 10$ | $0.584 \pm 0.013$ | $0.597 \pm 0.010$ | $0.087 \pm 0.002$ | $0.096 \pm 0.003$ | $0.179 \pm 0.009$ | $0.191 \pm 0.010$ |
| | $S = 100$ | $0.342 \pm 0.015$ | $0.353 \pm 0.016$ | $0.084 \pm 0.002$ | $0.093 \pm 0.003$ | $\mathbf{N/A}$ | $\mathbf{N/A}$ |
| PL-PG | $S = 1$ | $0.347 \pm 0.019$ | $0.356 \pm 0.020$ | $0.084 \pm 0.002$ | $0.093 \pm 0.002$ | $0.181 \pm 0.009$ | $0.194 \pm 0.009$ |
| | $S = 10$ | $0.345 \pm 0.021$ | $0.354 \pm 0.021$ | $0.084 \pm 0.002$ | $0.093 \pm 0.003$ | $0.180 \pm 0.009$ | $0.192 \pm 0.010$ |
| | $S = 100$ | $0.323 \pm 0.018$ | $0.332 \pm 0.018$ | $0.084 \pm 0.002$ | $0.093 \pm 0.003$ | $\mathbf{N/A}$ | $\mathbf{N/A}$ |

**Table 3:** Performance by time budget (10 minutes). Results obtained by the different algorithms while changing the slate size $K$ and the number of Monte Carlo samples $S$. All runs were completed in 10 minutes. We report in this table the average performance and its standard error over 6 different seeds. All statistics are reported on the validation set. The proposed algorithms give robust results in different settings.

| Algorithm | $S$ | MovieLens | | Twitch | | GoodReads | |
|---|---|---|---|---|---|---|---|
| | | $K = 5$ | $K = 100$ | $K = 5$ | $K = 100$ | $K = 5$ | $K = 100$ |
| LGP-MIPS | $S = 1$ | $0.645 \pm 0.011$ | $0.507 \pm 0.004$ | $\mathbf{0.098 \pm 0.004}$ | $0.107 \pm 0.005$ | $0.192 \pm 0.010$ | $0.205 \pm 0.011$ |
| | $S = 10$ | $0.528 \pm 0.012$ | $0.537 \pm 0.011$ | $0.086 \pm 0.003$ | $0.096 \pm 0.004$ | $0.179 \pm 0.007$ | $0.194 \pm 0.007$ |
| | $S = 100$ | $0.361 \pm 0.016$ | $0.374 \pm 0.017$ | $0.079 \pm 0.005$ | $0.088 \pm 0.006$ | $0.185 \pm 0.007$ | $0.198 \pm 0.006$ |
| LGP | $S = 1$ | $0.623 \pm 0.009$ | $0.646 \pm 0.004$ | $0.094 \pm 0.004$ | $0.108 \pm 0.005$ | $0.192 \pm 0.010$ | $0.207 \pm 0.011$ |
| | $S = 10$ | $0.513 \pm 0.012$ | $0.524 \pm 0.010$ | $0.086 \pm 0.003$ | $0.096 \pm 0.004$ | $0.179 \pm 0.007$ | $0.193 \pm 0.007$ |
| | $S = 100$ | $0.360 \pm 0.015$ | $0.372 \pm 0.016$ | $0.081 \pm 0.005$ | $0.088 \pm 0.005$ | $0.184 \pm 0.007$ | $0.197 \pm 0.006$ |
| PL-Rank | $S = 1$ | $\mathbf{0.750 \pm 0.006}$ | $\mathbf{0.779 \pm 0.003}$ | $\mathbf{0.102 \pm 0.004}$ | $\mathbf{0.113 \pm 0.005}$ | $0.188 \pm 0.010$ | $0.203 \pm 0.010$ |
| | $S = 10$ | $0.587 \pm 0.011$ | $0.600 \pm 0.009$ | $0.082 \pm 0.003$ | $0.092 \pm 0.003$ | $0.176 \pm 0.007$ | $0.190 \pm 0.007$ |
| | $S = 100$ | $0.332 \pm 0.016$ | $0.334 \pm 0.016$ | $0.081 \pm 0.005$ | $0.088 \pm 0.005$ | $\mathbf{N/A}$ | $\mathbf{N/A}$ |
| PL-PG | $S = 1$ | $0.342 \pm 0.017$ | $0.356 \pm 0.019$ | $0.084 \pm 0.004$ | $0.096 \pm 0.004$ | $0.181 \pm 0.010$ | $0.196 \pm 0.004$ |
| | $S = 10$ | $0.348 \pm 0.018$ | $0.355 \pm 0.020$ | $0.079 \pm 0.002$ | $0.090 \pm 0.003$ | $0.176 \pm 0.007$ | $0.190 \pm 0.007$ |
| | $S = 100$ | $0.305 \pm 0.018$ | $0.313 \pm 0.018$ | $0.081 \pm 0.005$ | $0.088 \pm 0.005$ | $\mathbf{N/A}$ | $\mathbf{N/A}$ |

**Table 4:** Performance by iteration. Results obtained with the different algorithms while changing the slate size $K$ and the number of Monte Carlo samples $S$. The number of iterations is decided by the slowest iterating algorithm (`PL-PG`) after running for 10 minutes. We report in this table the average performance and its standard error over 6 different seeds. `PL-Rank` gives the best results, followed by the `LGP` family.

### A.3.2 Effect of fixing $\beta$

In this section, we want to validate the intuition we built throughout the paper about the behaviours of both the Plackett-Luce and **LGP** policy classes. We focus on MovieLens (Harper & Konstan, 2015), a medium scale dataset that allows us to test all our methods, regardless of their potential to scale to harder problems. For all experiments in this section, we fix $L = 100 \ll P$ and we study the impact of fixing the action embeddings $\beta$. As discussed in Section 3, having the embeddings fixed is a natural solution to improve the learning of these large scale decision systems as it can reduce both the variance of the gradient estimates and the running time of the optimization procedure. To validate this, we focus on the Plackett-Luce policy and define two slightly different parametrizations:

- Learn $\theta$: this is the parametrization introduced in Equation (10), with $\beta$ fixed and we only learn the parameter $\theta$.

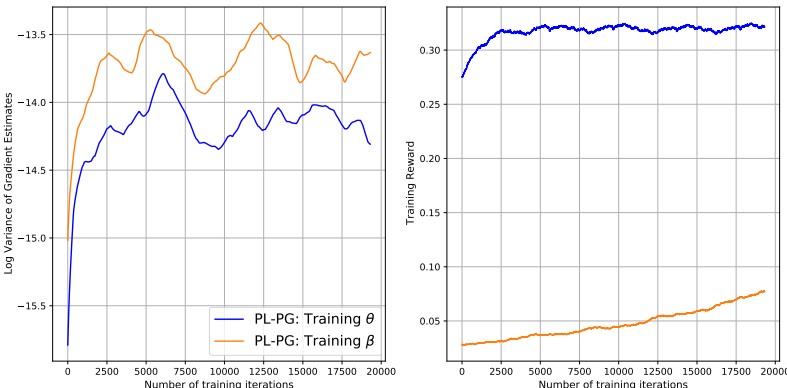

**Figure 3:** Experiments on the MovieLens dataset: We look at the effect of fixing the action embeddings $\beta$ on the training of Plackett-Luce policies. Training $\beta$ results in a slow optimization procedure and gradient estimates with bigger variance.

- Learn $\beta$: We treat $\beta$ as a parameter after initializing it with the SVD values. Because our user embedding function $h_\theta$ is linear in $\theta$, we get rid of this parameter as it becomes redundant once $\beta$ can be optimized. This gives the following parametrization:

$$\forall (x, a) \in \mathcal{X} \times \mathcal{A}, \quad f_\beta(a, x) = M(X)^T \beta_a$$

We train a Plackett-Luce policy with both parametrizations for one epoch, while fixing the slate size $K = 2$ and the Monte Carlo samples to $S = 1$. In this experiment, our objective is not to produce the best policies but to understand how fixing the embeddings can impact the optimization procedure. We report in Figure 3 the evolution of both the gradient estimate variance and the reward on the training data for both parametrizations. We can observe that treating $\beta$ as a parameter to optimize, even when initialized properly leads to slow learning. The variance of the gradient estimate when learning $\beta$ is bigger than the variance when learning $\theta$ with $\beta$ fixed, and this will get bigger for problems with larger action spaces as $P \gg L$. We suspect that this is one of the reasons that explain the pace of learning when optimizing $\beta$. The same phenomena was observed in Sakhi et al. (2023b). The same experiments demonstrated that training $\theta$ alone was **twice** as fast as training $\beta$ in this experiment. This suggests that fixing $\beta$ to a good value is beneficial for training large scale decision systems both in terms of iteration efficiency. We advocate for fixing the action embeddings $\beta$ when learning large scale MIPS systems.

### A.3.3 Impact of the slate size $K$.

One of the caveats of the Plackett-Luce slate policy is that its gradient estimate has a variance that grows with the slate size $K$, reducing its scope of applications to modest slate sizes. the gradient estimate of `LGP` however does not suffer from this issue, and we want to showcase that with a simple experiment. We derived gradient estimates for **LGP**-based methods that have a variance that scales in $\mathcal{O}(1/\sigma^2)$. Although the standard deviation $\sigma$ can be treated as a hyperparameter depending on the task, to allow a fair comparison of the gradient variance of these methods, we set $\sigma$ to a particular value coming from the following observation. For a particular action $a$:

$$h \sim \mathcal{N}(h_\theta(x), \sigma^2 I_L) \implies h^T \beta_a \sim \mathcal{N}(h_\theta(x)^T \beta_a, \sigma^2 ||\beta_a||^2)$$
$$\implies h^T \beta_a = h_\theta(x)^T \beta_a + \epsilon_a$$

with $\epsilon_a \sim \mathcal{N}(0, (\sigma ||\beta_a||)^2)$. This can be interpreted as adding a scaled guassian noise to the score of action $a$. As we add standardized Gumbel noise $\gamma_a \sim \mathcal{G}(0, 1)$ to the action scores to define the Plackett-Luce policy, we want, for a fair comparison, to have:

$$\forall a \in \mathcal{A}, \quad \sigma ||\beta_a|| \approx 1$$

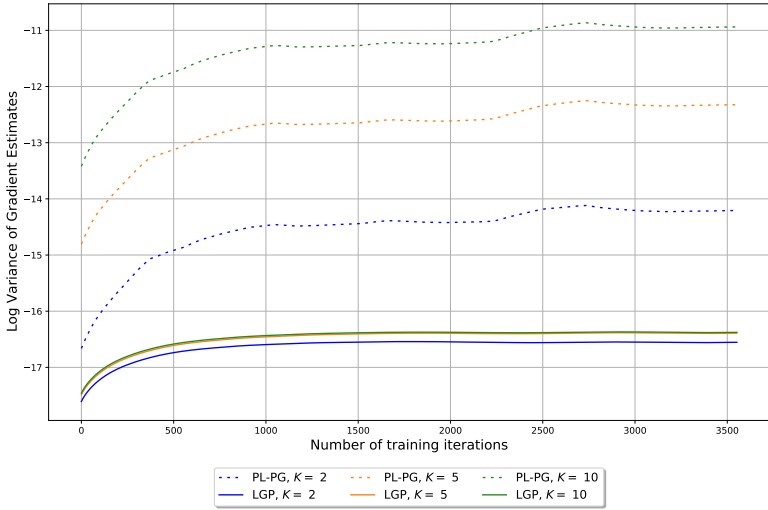

**Figure 4:** Impact of the slate size $K$ on the log variance of the gradient estimate of both `PL-PG` and `LGP` on MovieLens. Contrary to `LGP`, `PL-PG` has a gradient estimate with a variance that grows with $K$.

One heuristic to approximately achieve that is to compute the empirical mean of the $\beta$ norms $B = \frac{1}{P} \sum_{a \in \mathcal{A}} ||\beta_a||$ and set $\sigma = 1/B$. This value will be used for this experiment.

We use the MovieLens dataset, and train the `LGP` policy *without exploiting the MIPS index on $\beta$*, and Plackett-Luce with the parametrization of Equation (10) for 10 epochs with $S = 1$, while varying the slate size $K \in \{2, 5, 10\}$. Note that all policies are initialized with the same random seed for a fair comparison. We report the evolution of the variance of the gradient estimate alongside the reward on the training data. The results of these experiments are presented in Figure 4.

Focusing on the evolution of the variance, we can see that Plackett-Luce does indeed have a variance that grows with $K$ contrary to **LGP** that has a variance of its gradient estimate staying at the same scale no matter the value of $K$. We argue that this has a direct impact on the optimization procedure as for the same value of $K$, we observe in Figure 1 that **LGP**-based methods outperform Plackett-Luce learning schemes consistently, making it a good candidate for learning slate policies.

### A.3.4 Experiments with Neural Networks.

For these experiments, we want to explore deep policies and see if we can still empirically validate our findings in this case as well. For this, we adopt the following function to compute the user embedding $h_{\theta_1, \theta_2}$:

$$h_{\theta_1, \theta_2}(X) = \texttt{sigmoid}\left(M(X)^T \theta_1\right) \theta_2, \tag{13}$$

which boils down to a sigmoid, two layer feed forward neural network with both $\theta_1$ and $\theta_2$ of size $[L, L]$. We run `PL-PG`, `PL-Rank` and `LGP-MIPS` on the three datasets, for the same running time (60 minutes) and cross-validate the learning rate choosing the best value for each algorithm. We aggregate the results on Figure 5. The plot suggests that even for deep policies, `LGP-MIPS` outperforms **Plackett-Luce**-based methods on all datasets and for different values of the number of Monte Carlo samples $S$. We also observe that training in this case is more unstable, especially for **Plackett-Luce**-based methods as having more parameters accentuate the variance problems of their gradient estimates. It is noteworthy that, the reward obtained with deep policies in our experiment is less than the one achieved by linear policies. This suggests that deep policies require additional care when training, and we might want to stick to simple policies if we are interested in fast and reliable optimization.

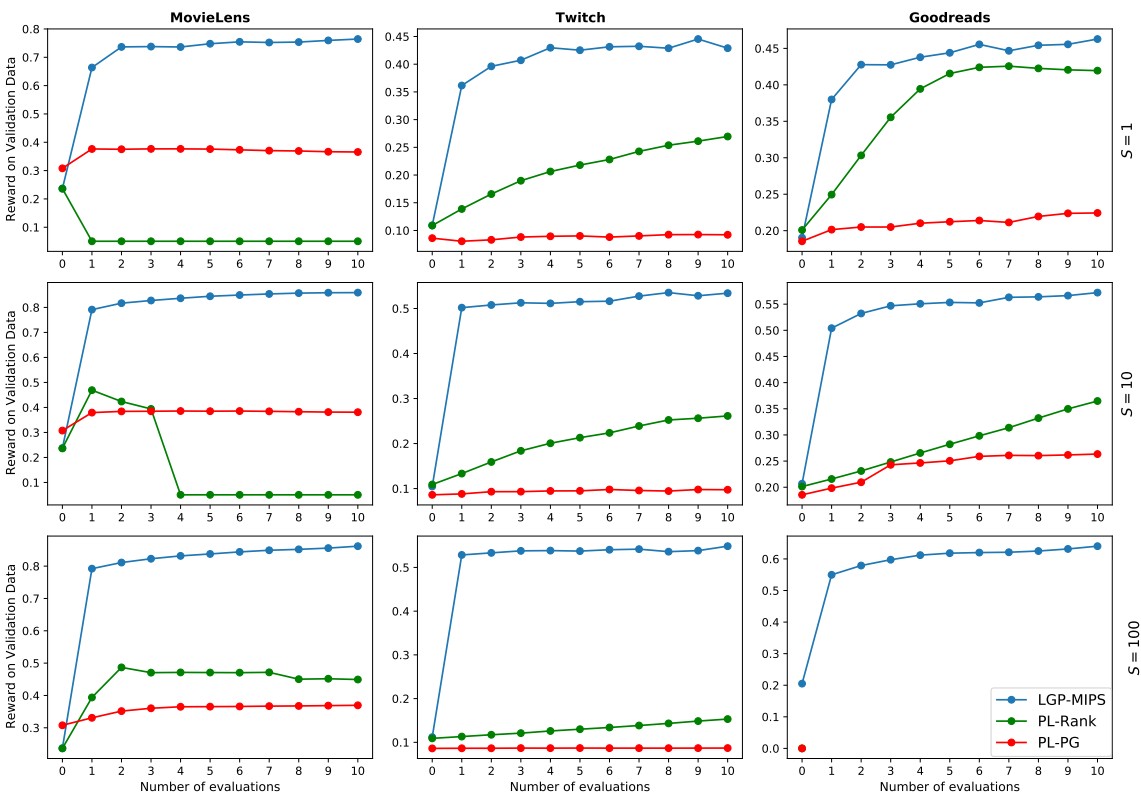

**Figure 5:** The performance of slate decision functions, with Neural Network Backbones, obtained by our different training algorithms after running the optimization for 60 minutes.

