# OpenReview forum: "Fast Slate Policy Optimization: Going Beyond Plackett-Luce"
_TMLR — Accepted by TMLR_

### Review · Reviewer_PZmb · 2023-09-04

**Summary Of Contributions:**

The paper studies the problem of slate recommendation via policy learning and proposes Latent Gaussian perturbation policies as replacement for the commonly used Plackett-Luce policies, thus addressing some of the shortcomings of the Plackett-Luce policies (high computational cost and high variance of gradient estimator).

**Audience:**

Yes

**Claims And Evidence:**

No

**Requested Changes:**

I think the paper needs to clearly show the proposed method’s benefits in an extensive empirical evaluation (the various runtime discussions are nice, but in the end the authors provide only asymptotic *upper* bounds). To that end, at a minimum, I would like to see
*) plots with average results + confidence intervals / standard deviations (and an explanation how these are obtained)
*) a better explanation why PL-PG does not work in the considered setting (what happens if one further increases S or the batch size or decreases the learning rate?)
*) performance as a function of the number of epochs rather than runtime – please also provide some details how runtime is measured
*) experiments with different values of K
*) a more thorough investigation of the variance of the different gradient estimators, discussed in the main part of the paper
*) plots that show the expected reward under the policy (the optimization objective)

**Strengths And Weaknesses:**

Strengths:
*) The proposed Latent Gaussian perturbation policy is conceptually simple and can be argued to have several advantages over the common Plackett-Luce policy. In the experiments, the proposed algorithms clearly outperform existing ones.

Weaknesses:
*) The empirical evaluation is rather limited (see my requested changes below); in particular, the experiments do not seem to report average results and do not provide any standard deviations
*) The paper is cumbersome to read, partly because it tries to introduce the most general concept but then only considers a special case (e.g., general decision functions -> parametric relevance functions -> parametric relevance functions with fixed action embeddings, or latent random perturbation -> latent Gaussian perturbation). I would recommend to shorten the presentation and refrain from providing the general description where only the concrete description is needed in order to make the paper more comprehensible and use the space for additional experiments.

There are a number of typos / minor issues:
*) Please refer to appendix sections rather than just saying “can be found in the Appendix”
*) Section 3, first sentence of second paragraph: some word is missing
*) Page 8: making the evaluation of ITS THE log density gradient easy
*) Page 9, first line: start sentence with “The” rather than “the”
*) Section 5, paragraph on Learning to Rank: remove white space before the period; later a period is missing
*) Page 8, last line: the notation UI for the user-item interaction session is unclear and one might confuse it for a product of U and I.
*) Page 18, last sentence: on the task, TO allow

---

> ### Author Response · Authors · 2023-10-09
> **Response**
>
> - "I think the paper needs to clearly show the proposed method’s benefits in an extensive empirical evaluation (the various runtime discussions are nice, but in the end the authors provide only asymptotic upper bounds). To that end, at a minimum, I would like to see *) plots with average results + confidence intervals / standard deviations (and an explanation how these are obtained) *) a better explanation why PL-PG does not work in the considered setting (what happens if one further increases S or the batch size or decreases the learning rate?) *) performance as a function of the number of epochs rather than runtime – please also provide some details how runtime is measured *) experiments with different values of K *) a more thorough investigation of the variance of the different gradient estimators, discussed in the main part of the paper *) plots that show the expected reward under the policy (the optimization objective)"
>
>
>
> Thank you for identifying some typos – we have corrected the document.
>
>
>
> Extensive experiments benchmarking the method have been added to the paper (see Figure 2 and Table 3, Table 4).  Figure 2 compares the algorithm efficiency by iteration.  Table 3 and Table 4 compare the algorithm performance for fixed time and iteration budgets respectively.  The superiority of LGP to PL-PG is clear over a range of slate sizes and Monte Carlo samples- especially on fixed time budgets (and large catalogue sizes) because of LGPs lower iteration cost.  The lower memory footprint of LGP is also evident in cases where the slate size and number of Monte Carlo samples makes it impossible to run PL-PG.
>
>
> We understand there is value in re-running experiments with multiple seeds and error analyses.  In the case of LGP compared to PL-PG the improvements in speed are so great that intra-seed variation is very small compared to the observed performance differences.

---

### Review · Reviewer_Tht2 · 2023-09-13

**Summary Of Contributions:**

This paper provides a novel algorithm for slates learning. The advantages of the proposed techniques, namely LGP, are that they show a better time complexity in terms of the size of the action space P. The authors provide empirical evidence that the propose method is also providing better reward than other state-of-the-art available options.

**Audience:**

Yes

**Broader Impact Concerns:**

I do not foresee any ethical concerns about the paper.

**Claims And Evidence:**

No

**Requested Changes:**

See weaknesses for details-. The main changes are related to the structure and writing of the paper to have a clear picture of the contribution and to provide formal statements for the claims requiring formal proof.

**Strengths And Weaknesses:**

The topic is interesting, and the method is sound. My opinion is that the paper is not well structured, and this makes its reading hard. For instance, the problem formulation is entwined with the introduction.

Moreover, I think the authors should put more effort into stating explicitly when they are presenting something from the literature and when it is a novel contribution. For instance, section 2 is all about the PL policies, which are part of the literature; therefore, I think using 1.5 pages is not focusing the paper on the most interesting part.

Finally, I think the paper lacks some formalism in stating the properties of the developed method. For instance, I would have expected some formal theorem/lemmas showing the properties of what the authors proposed. At the same time, they only provide informal statements and defer the discussion to the appendix. I think you should put some effort into reformulating such statements in a formal way.

Minor:
- "In this paper we demonstrate such an algorithm". Not clear
- The introduction proposes both a description of the problem and the problem formulation. I suggest you to reorganize such a section to highlight the two elements.
- you need to define \nu(\mathcal{X}) in equation before Eq. (3)
- add punctuation to formulas
- "it induces properties that are hard to deal with:" Please explain
- add a textual description of the pseudocode

---

> ### Author Response · Authors · 2023-10-09
> **Response**
>
> - “The topic is interesting, and the method is sound. My opinion is that the paper is not well-structured, and this makes its reading hard. For instance, the problem formulation is entwined with the introduction.“
>
>
> Thank you for identifying some typos – we have corrected the document.
>
>
> The introduction does indeed move quickly to the use of mathematical notation to formalize the problem early. We think the formality is necessary to clarify the problem, as we are unaware of any exposition outlining argsort algorithms based on Placket-Luce policies, and employing both the Reinforce and maximum inner product search (MIPS).  We understand that this is not to everyone’s taste and welcome precise suggestions to improve the structure.
>
>
>
> - “Moreover, I think the authors should put more effort into stating explicitly when they are presenting something from the literature and when it is a novel contribution. For instance, section 2 is all about the PL policies, which are part of the literature; therefore, I think using 1.5 pages is not focusing the paper on the most interesting part.”
>
>
>
> Section 2 is indeed literature review, and, also provides some useful commentary on the current state of the literature explaining in detail:
>
> - Restrictions required to exploit maximum inner product search algorithms
>
> - Coupling of Placket-Luce with Reinforce
>
> - Iteration cost of these algorithms, and proposals to reduce this (and limitations)
>
> - Variance of the Reinforce algorithm – particularly with respect to how the Variance behaves with increased slate size.
>
> As these limitations are quite technical and are not outlined clearly elsewhere, we feel it is justified to use 1.5 pages for this purpose. Within our structure, Section 3 can now offer solutions. The last paragraph of the introduction also notes that Section 2 is review and Section 3 are new contributions.  We are open to further suggestions to improve clarity and acknowledge that this structure may not be to everybody’s taste. We were pleased to find that Reviewer 6C5d actually found the paper easy to read.
>
>
> - “Finally, I think the paper lacks some formalism in stating the properties of the developed method. For instance, I would have expected some formal theorem/lemmas showing the properties of what the authors proposed. At the same time, they only provide informal statements and defer the discussion to the appendix. I think you should put some effort into reformulating such statements in a formal way.”
>
>
> Indeed, the intuition that the variance of the PL-PG grows linearly with the slate size (K) requires that the covariance terms do not cancel (variance formula given on page 5).  While this intuition is backed up by empirical observations (see Figure 4 of the revised manuscript).  A proof that the covariance terms do not cancel would be a valuable contribution, but this is beyond the scope of this paper.

---

### Review · Reviewer_6C5d · 2023-09-14

**Summary Of Contributions:**

The paper studied slate recommendation in offline contextual bandit setting. Previous works commonly optimize the ranking policy under Plackett-Luce distribution. The authors aim to solve the issues with Plackett-Luce policy class such as slow in computation and suffering large variance. The authors proposed Latent Gaussian Perturbation (LGP) with the key idea of relaxing the objective to be differentiable by perturbing in the latent space instead of in the action space. LGP reduces computation complexity and enjoys smaller variance in gradient estiamtes. Empirical results on real-world datasets validated the effectiveness of proposed method.

**Audience:**

Yes

**Claims And Evidence:**

Yes

**Requested Changes:**

Please see weaknesses.

**Strengths And Weaknesses:**

Strengths

1. The problem is well-motivated and practical.

2. The idea of latent space perturbation is intuitive. The derivation of the gradient estimates discussion of its advantage is sound and insightful.

3. The improvement in experiments under same training time is significant.

4. The writing is clear and easy to follow.


Weaknesses

No major weaknesses were identified. I have some minor concerns:

1. I would suggest the authors make formal statements regarding statistical improvement of LGP. 1) Is the gradient estimates unbiased? I noticed related discussion under "Well behaved gradient". The authors may consider using a theorem to formally make the claim.  2) Similar question regarding "Better variance". Could the authors provide a formal statement (and proofs) to compare the variance of gradient estimates between LGP and baseline?

2. In the experiments, the authors evaluate LGP and baselines under the same training time (60minutes) and showed the effectiveness in accelerating computation. I would suggest the authors also report results under the same training iterations to validate the impact of smaller  variance.

---

> ### Author Response · Authors · 2023-10-09
> **Response**
>
> - “I would suggest the authors make formal statements regarding statistical improvement of LGP. 1) Is the gradient estimates unbiased? I noticed related discussion under "Well behaved gradient". The authors may consider using a theorem to formally make the claim. 2) Similar question regarding "Better variance". Could the authors provide a formal statement (and proofs) to compare the variance of gradient estimates between LGP and baseline?”
>
>
>
> The estimation of the gradient (Equation (9)) is unbiased. This property improves on the covariance gradient proposed in Sakhi (2023), which requires self-normalized importance sampling to be estimated (biased, but consistent).  unaccelerated approximations of the gradients, such as PL-PG or PL-Rank estimators are also unbiased, but either suffer from large variance or exploit an additional assumption on the reward to reduce their variance.
>
>
>
> The variance formula given on page 5 is strongly suggestive that the variance of reinforce under Placket-Luce policies increases linearly with the slate size.  This intuition is supported by the empirical results on the gradient of the variance in Figure 4 (revised document).  In contrast the variance of LGP does not depend on the slate size.  We agree that it would be a useful contribution to prove that the variance of PL-PG always increases linearly with the slate size, as this would demonstrate that for sufficiently large slate size LGP will outperform PL-PG.  Completing this proof would require showing that the covariance terms never cancel and is beyond the scope of this paper.
>
>
>
> - “In the experiments, the authors evaluate LGP and baselines under the same training time (60minutes) and showed the effectiveness in accelerating computation. I would suggest the authors also report results under the same training iterations to validate the impact of smaller variance.”
>
>
>
> A plot showing the algorithm effectiveness per iteration is shown in Figure 2 in the updated document.

---

### Author Response · Authors · 2023-10-09
**General Response**

We thank the reviewers for their thoughtful comments, and apologies for the delay in responding.  The first author of the paper had a very big deadline concurrently.  The paper has been updated to incorporate some of the changes requested.  Briefly:

- Two of the referees requested more extensive experiments, these have been added to the paper.

- Precise suggestions made for improving the writing were incorporated.

- We agree with referees that theoretical results may be able to prove conclusively that LGP outperforms PL-PG, but we consider these theoretical investigations to be beyond the scope of this paper.

- The referees disagreed on the readability of our paper.  We understand the concerns of referee Tht2, but we think there are advantages to our rather extensive commentary on the weaknesses of the state of the art before providing our solution.

---

### Decision · Action_Editor_b8VV · 2023-11-01

**Recommendation:** Accept with minor revision

**Comment:**

The reviewers appreciated the considered problem and also the proposed approach. However, initially, there were concerns regarding experimental evaluation and also presentation of the paper. The concerns regarding experiments have been partly addressed by adding new experimental results but these results should also be carefully discussed in the main paper and not only in the appendix - in this regard the paper needs to be improved. Regarding presentation, two reviewers raised concerns in their initial reviews but were also not fully satisfied with the changes made by the authors. The authors implemented several changes addressing parts of the concerns but avoided making bigger adjustments to, for instance, make the paper more "comprehensible" (PZmb) by focussing the presentation more on the actually addressed aspects. I also saw the authors' response in that regard and would suggest to only introduce the directly relevant parts first and later have a more general discussion as needed. Considering all this, I am recommending a minor revision addressing the open concerns of the reviewers.
Side note: The reviewers also suggested providing theoretical statements/insights but I agree with the authors that these are not necessary for accepting the paper and might not be easy to derive.

**Audience:**

Yes, the considered problem is relevant itself and also relevant to many applications relevant to TMLR's audience, e.g., recommender systems.

**Claims And Evidence:**

Yes, the main claim is about the scalability of a proposed approach for computing slates. The runtime efficiency of the proposed method is demonstrated in sufficiently many experiments in comparison to relevant baselines. There are some statements that are not formally proven but supported by empirical evidence which I consider sufficient.